# Using Electric Stimulation of the Spinal Muscles and Electromyography during Motor Tasks for Evaluation of the Role in Development and Progression of Adolescent Idiopathic Scoliosis

**DOI:** 10.3390/jcm13061758

**Published:** 2024-03-19

**Authors:** Christian Wong, Hamed Shayestehpour, Christos Koutras, Benny Dahl, Miguel A. Otaduy, John Rasmussen, Jesper Bencke

**Affiliations:** 1Department of Orthopedic Surgery, Copenhagen University Hospital, 2650 Hvidovre, Denmark; jesper.bencke@regionh.dk; 2Department of Orthopedic Surgery, Rigshospitalet, 2100 Osterbroo, Denmark; benny.dahl@regionh.dk; 3Department of Materials and Production, Aalborg University, 9220 Aalborg, Denmark; hs@anybodytech.com (H.S.); jr@mp.aau.dk (J.R.); 4Department of Computer Science, Universidad Rey Juan Carlos, 28933 Madrid, Spain; christos.koutras@urjc.es (C.K.); miguel.otaduy@urjc.es (M.A.O.)

**Keywords:** electric stimulation, spine muscle, Cobb angle, adolescent idiopathic scoliosis, AIS, EMG, electromyography

## Abstract

**Introduction**: The role of the spinal muscles in scoliogenesis is not fully substantiated. Do they act scoliogenic (inducing scoliosis) or counteract scoliosis in adolescent idiopathic scoliosis (AIS)? In this study, we will examine this by using selectively placed Transcutaneous Electric Stimulation (TES) combined with a cinematic radiographic technique and by performing electromyographic (EMG) evaluations during various motor tasks. **Method**: This is a cross-sectional study of subjects with small-curve AIS. Using cinematic radiography, they were evaluated dynamically either under electrical stimulation or when performing motor tasks of left and right lateral bending and rotation whilst measuring the muscle activity by EMG. **Results**: Forty-five patients with AIS were included as subjects. Five subjects volunteered for TES and six subjects performed the motor tasks with EMG. At the initial visual evaluation, and when stimulated with TES, the frontal plane spatial positions of the vertebral bodies showed discrete changes without an apparent pattern. However, analyzing the spatial positions when calibrated, we found that the spinal muscles exert a compressive ‘response’ with a minor change in the Cobb angle (CA) in small-curve AIS (CA = 10–20°). In larger curves (CA > 20°), TES induced a ‘larger deformity’ with a relative four-fold change in the CA compared to small-curve AIS with a ratio of 0.6. When evaluating local amplitude (peak) or cumulative (mean) EMG signals, we were unable to find consistent asymmetries. However, one subject had rapid progression and one regressed to a straight spine. When adding the absolute EMG ratios for all four motor tasks, the subject with progression had almost 10-fold less summed EMG ratios, and the subject with regression had more than 3-fold higher summed EMG ratios. **Discussion**: Based on these findings, we suggest that the spinal muscles in small-curve AIS have a stabilizing function maintaining a straight spine and keeping it in the midline. When deformities are larger (CA > 20°), the spine muscle curve exerts a scoliogenic ‘response’. This suggests that the role of the muscles converts from counteracting AIS and stabilizing the spine to being scoliogenic for a CA of more than 20°. Moreover, we interpret higher EMG ratios as heightened asymmetric spinal muscle activity when the spinal muscles try to balance the spine to maintain or correct the deformity. When progression occurs, this is preceded or accompanied by decreased EMG ratios. These findings must be substantiated by larger studies.

## 1. Introduction

Adolescent Idiopathic Scoliosis (AIS) is a spinal deformity defined by a coronal plane curvature with rotation [1]. AIS may progress rapidly in periods of growth [2]. Thus, predictions of curve progression during or before this period would be purposeful but are challenged by the multifactorial etiology and unpredictable nature of curve progression [3]. Major radiological predictors are the curve magnitude and type, the curve flexibility and thoracic curve correction by bracing. Other less sensitive predictors are apical vertebral rotation, rib morphology and pelvic tilt [4]. Prediction is important, since larger curves might progress, and the adolescent with AIS may suffer from back pain and poorer quality of life in adulthood [5,6]. Experimental techniques may allow us to identify factors for prediction by clarifying relationships of underlying disease-related mechanisms [3]. Today, the etiology of AIS is still elusive and considered to be caused by a wide range of factors of biomechanical and genetic nature [7,8,9]. Wong [10] advocated a common muscle-dependent mechanical mediated pathway by rotational instability in the period of adolescent growth leading to AIS, and Castelein and co-workers described that AIS is a mechanical consequence of preexistent rotation in the normal spine, direction of dorsal shear forces by the human upright position and normal adolescent growth [10,11,12,13]. Modi et al. [14] observed a “wavy” pattern of the deformity in small-curve AIS and suggested a “balancing/tuning” mechanism of spinal muscle adaptation determining the course of spontaneous progression, remaining stable or demonstrating regression [14]. The majority of AIS remains stable but might progress or regress [15]. An imbalance in the spinal muscles is suggested to play an integral role in the initial development and progression of AIS [10,16,17,18]. Differences in the spinal muscles on the concave and convex side in the muscle cross-sectional area and muscle fibre type examined by MRI, motion, strength and electromyographic responses to motor tasks indicate the existence of such a muscular imbalance [19,20,21,22,23,24,25,26,27,28]. The stabilizing role of the spinal muscles is still not fully understood [16,17,21,23,29,30,31,32]. Grivas et al. [29] speculated whether the spinal muscles act scoliogenic (inducing scoliosis) or counteract scoliosis and suggested this hypothesis be tested by electrical stimulation.

### 1.1. Electromyography (EMG)

The described differences and imbalance in the spinal muscles have been suggested to be related to differences in muscle activation, causing localized asymmetry in EMG responses [28]. Studies have examined asymmetries in surface EMG comparing various paravertebral regions around the (primary) deformity and examining EMG onset, cumulative signals and amplitude in normalized and non-normalized EMG. This has been evaluated for AIS, compared to typically developed adolescents, in progressive compared to non-progressive AIS, for different curves, different types of scoliosis (i.e., neuromuscular), and for various tasks (static postural, isometric, isokinetic, or mechanical perturbations) and suggested to be evaluated as ratios using the following formulae:EMG convex/EMG concave, denominated X/Y(1)
EMG convex/(EMG convex + EMG concave), denominated X/X + Y(2)
where the EMG signal might be either the amplitude (peak) or cumulative (mean) signal, and where a symmetrical EMG signal would evaluate close to 1 for (1) and close to 0.50 for (2) [28]. Asymmetry in these ratios of EMG has been assumed to predict the risk of progression, but EMG studies have shown diversity without a clear pattern of asymmetry. However, moderate to larger EMG asymmetries in a progressive thoracic single curve, during static postural tasks, and at the apex and in lower regions of the scoliosis curve have been demonstrated [28]. In this study, we evaluate EMG asymmetries in various motor tasks to investigate possible deformity prediction in AIS.

### 1.2. Transcutanous Electric Stimulation

Functional electrical stimulation for treatment demonstrated short-term correction in the 1980s but without a lasting effect [33,34,35,36,37,38]. Although activating musculature with TES was abandoned as treatment, experimental studies have examined changes in lumbar stabilizing muscles when applying TES using ultrasonic visual detection of muscles [39,40,41], and examination of TES for scoliogenicy has been suggested [29]. For this purpose, we will introduce selectively placed TES in AIS combined with the cinematic low-dose radiographic technique [42] to evaluate the motion response in the vertebral bodies in spines with AIS to explore the spinal muscle’s role in scoliogenicy [29].

### 1.3. The Current Standard Practices for AIS Prediction and Treatment

The current standard practices for predicting and managing AIS focus on early detection, monitoring, and intervention based on the severity and progression risk of the curvature. For mild curves, observation and regular monitoring of the spine’s curvature to track curve progression is common with periodic clinical evaluations and radiographic assessments, where the risk of progression can be evaluated by a combination of serial X-rays, menarche history, serial height measurement, triradiate cartilage status, Risser grade, and Sanders skeletal maturation staging. In moderate curves in a growing adolescent, bracing is often utilized. Both observation and bracing can be combined with physiotherapy and exercises, such as the Schroth method. Surgical interventions are offered in severe cases of AIS when rapidly progressing or causing significant problems such as back pain.

### 1.4. The Role of Muscle Imbalance in the Development and Progression of AIS

Muscle imbalances are hypothesized to influence the development and progression of AIS [10,14,16], primarily involving the muscles that provide rotational stability to the thoracic and lumbar regions, as well as specific muscles such as the psoas and quadratus lumborum. Wong [10] suggested spinal muscles with thoracic and lumbar rotational functions, thus maintaining ‘rotational’ stability to those areas, are key determinants in a ‘common pathway of imbalance’ in AIS and loss of function of those muscles would facilitate AIS. Previous publications by Wong et al. [29] and Griva et al. [32] showed that an intervention targeting the psoas muscle by injections of botulinum toxin resulted in improvements in the curvature and rotation of the spine, and radiological evaluation of asymmetry of the 12th rib length suggested that the quadratus lumborum muscle could affect lateral lumbar curves in AIS.

## 2. Material and Methods

The Regional Committee on Health Research Ethics, RegionH, Copenhagen, Denmark approved the studies (H-17034237). We obtained oral and written consent from the caregivers and the older subjects (when appropriate), and the study was conducted according to national guidelines and the Declaration of Helsinki. The study was investigator-driven and supported by the Department of Orthopedics, Hvidovre Hospital, Hvidovre, Denmark.

### 2.1. Design of the Study

The study was a cross-sectional study of patients with AIS. The patients were included among subjects from the outpatient clinic at our hospital as a convenience sample. The inclusion criteria were adolescents diagnosed with a mild to moderate degree of scoliosis with Cobb angles between 10° and 30°. The subjects were informed by an experienced pediatric orthopedic surgeon in close consultation with the primary caregivers to ensure that the subjects were able to understand, cooperate, comply or cope with the procedures of the study. They were excluded if they participated in other studies/had treatments affecting their spine or spinal muscles or had spine-related operative treatment. The subject wore shorts and a specially constructed t-shirt, which was open in the back to expose the spinal column whilst covering the chest at the front. When evaluated radiographically, this substituted an already planned follow-up to minimize the radiation dose. The subjects were followed medium term as part of the clinical follow-up either until maturity with observation with or without interventions of physiotherapy or bracing. If spinal surgery was indicated, they were excluded from the follow-up. We retrieved baseline data of age, height, body weight, other spine-related metadata, MRI findings when available, information on spine-related treatments as well as the radiological medium-term follow-up. Radiological examinations were performed approximately every six months from detection to maturity according to clinical guidelines [43].

### 2.2. Radiological Evaluation

During electric stimulation, a ‘radiograph’ in the anterioposterior projection of the subject was performed. The subject was standing facing the image intensifier/detector with extended hips and knees and with the feet 10 cm apart. Two lead aprons were placed in the interscapular (mammary) and sacral (genitals) regions. The distance to the radiation tube (source to detector) was 230 cm. One ‘capture’ was needed to record the spinal deformity. The examination was performed using a digital DelftDI D2RS system with automatic exposure control. The following acquisition parameters were utilized: tube potential (71 kVp, density set at 0), with grid, using automated mA selection exposure control and additional filtration (1 mm Al + 0.1 mm Cu). The pixel size was 0.139 mm with a maximum resolution of 3.6 lp/mm. A radiopaque ruler was placed adjacent to the subject. The ‘radiographic capture’ entailed a consecutive series of 4–8 continuous radiographs, creating a cinematic sequence, capturing the short dynamic movement of the spine. The radiation dose was eight-fold lower than a standard radiological examination for AIS [42].

### 2.3. Muscle Activity Evaluated by EMG in Various Motor Tasks

The subjects performed the four motor tasks of left and right lateral bending and left and right rotation in a motion capture laboratory with EMG monitoring [44]. These tasks reflected specific tasks of everyday life and were chosen to not impose unnecessarily high loads on the subjects [28,45]. The tasks were carried out on a flat linoleum floor and performed with five repetitions as described in detail in the Appendix A. To ensure safety and to familiarize the subjects with the tasks, a series of test tasks were carried out before trial initiation, thus minimizing the risk of accidents and avoiding excess stress. Before trial initiation, the subjects had reflective markers installed whilst standing still in the middle of the room and while the cameras were filming. The subjects were placed in the ‘portrait’ mode of anatomical stance at rest. The position was similar to the position in the radiological examination. The duration of the experimental procedure was 45 min including the application of the reflective markers.

#### 2.3.1. Motion Capture

We placed 43 reflective markers with tape on the skin over various bone projections following the marker protocol illustrated in Figure 1.

Marker trajectories were recorded by an eight-camera motion analysis system with a frame rate of 100-Hz (T40 cameras, Vicon motion Systems Ltd., Oxford, UK); thus, the 3-dimensional coordinates of the markers determined the movements whilst performing the motor tasks.

#### 2.3.2. EMG

Ten Myon (Cometa, Milan, Italy) electrodes were placed on the back and tailored to the primary curve of the AIS as illustrated in Figure 1. Before placement, the skin was cleaned with alcohol and abraded according to SENIAM standards [46]. The Wireless Myon System (Cometa, Milan, Italy) recorded the muscle activity of the different parts of the left and right back extensor muscles whilst performing motor tasks using a sampling frequency of 1000 Hz. The raw EMG data were rectified and filtered using a high-pass filter with a cutoff frequency of 20 Hz and subsequently smoothed using a rolling-window RMS of 500 ms with 1 ms steps. Further, the post-processing of the data included evaluating all the video sequences for the correct execution of the motion task. This was evaluated by one rater. The start of each repetition of the motion task was defined when the ipsilateral acromion marker initiated the movement in either the caudal direction (at lateral flexion tasks) or in front of the contralateral acromion marker (rotation task). EMG outcome measures were extracted using a custom-written Matlab (Version 10.0) script calculating the peak value and mean non-normalized EMG signal for the upper, mid-curve (apex) and lower paravertebral region of the primary curve during each repetition of the motion task [28]. Finally, the ratios were computed according to Equations (1) and (2). In post-processing using Excel (Microsoft 365, Version 2402), we calculated the normalized ratios to the average EMG ratio of the subject. The differences for the upper, apex and lower regions were evaluated as well as the summated difference and absolute difference for the three regions. Examples of the EMG data are displayed in the Appendix A.

### 2.4. Electric Stimulation of the Spinal Muscles

The subject was standing facing the radiation tube in the anterioposterior projection whilst performing one cinematic capture of the AIS. The spinal muscles were stimulated transcutaneously, either unilaterally or bilaterally, laterally to process spinosis at predetermined positions relative to the primary curve of the AIS. Figure in Section 3.2 displays the placement of the stimulation electrodes. We utilized a CefarCompex electric stimulator Mi-Theta 500 Chattanooga Physio (Programming mode, NMES, Cont. Conventional) [47]. The stimulation intensity was manually controlled with a gradual increase in intensity until a visible muscle response was detected (strength: 20–80 mA, pulse duration: 200 μS, frequency: 30 Hz) whilst monitoring the subject’s well-being. The procedure was interrupted if the subject either refused to participate or if the procedure became too painful. All involved investigators had electric stimulation performed on themselves before testing the subjects. When evaluating the cinematic changes during stimulation, we measured the calibrated distance between the upper and lower endplate of the Cobb angle. This was performed for the initial unstimulated radiographic capture and at maximal stimulation (maximal change in Cobb angle during stimulation). We calculated the following:R_dis_ = (d_u_ − d_s_)/d_u_(3)
R_ang_ = (α_u_ − α_s_)/α_u_(4)
R_n_ = R_ang_/R_dis_(5)
where R_dis_ (3) is the normalized change in distance between the upper and lower vertebral endplate when stimulated (resulting in compression) d_s_ and the unstimulated distance d_u_, and R_ang_ (4) is the Cobb angle change by stimulation α_s_, normalized by the unstimulated Cobb angle α_u_. The R_n_ (5) is the ratio between the two former values.

### 2.5. Data Analysis

Post-processing and synchronizing of the video capture and EMG were performed using Matlab version 10.0. Evaluation for asymmetries of the EMG data was performed using Microsoft^®^ Excel^®^ for Microsoft 365 MSO (Version 2202 Build 16.0.14931.20942/32-bit). Video capturing was performed using Snagit, version 2023.0.1. We evaluated the radiological cinematic sequences during electric stimulation using Synedra View Personal view 22.0.0.1 for measuring the non-calibrated and calibrated distances and Cobb angles. Statistical analyses were performed using IBM SPSS Statistics, version 25 (IBM, Richmond, VA, USA) using descriptive statistics for determining medians and quartiles of the age and Cobb angle. Further statistical analyses for significance were not carried out due to the low sample size, thereby limiting the accuracy and generalizability of study results with reduced reliability, a higher margin of error, lower statistical power, increased variability, risk of bias and higher risk of type I and II errors.

## 3. Results

Patients with AIS from our pediatric orthopedic outpatient clinic were screened for eligibility. Forty-five patients with their caregivers were included as subjects after receiving oral and formal written information from an experienced pediatric orthopedic surgeon using the national guidelines for informed consent and ensuring the subject and their caregivers understood the study purpose, procedures, risks, and benefits. After the caregivers accepted participation, their medical records were screened according to the inclusion and exclusion criteria. If still eligible, the subjects were scheduled to have electric stimulation performed at their next planned radiological follow-up and performed motion capture and EMG recording at the department’s gait laboratory. Figure 2 shows a flow chart of the history of the subject’s participation and exclusion.

### 3.1. EMG Ratio

Nineteen patients with AIS were recruited as subjects. All performed the four tasks of left and right lateral bending and rotation, respectively. Six subjects performed all repetitions of the four tasks adequately, thus thirteen were excluded. The gender ratio (female:male) was 6:0 and the median age was 14.37 years of age (range: 7.0, Q1: 11.97, Q3: 15.97). The median Cobb angle at examination was 18.0 (range: 57, Q1: 11.25, Q3: 33.75). The final median Cobb angle was 15.0 (range: 21, Q1: 5.6, Q3: 22.0). Figure 3 displays the summed absolute values of the mean and peak EMG for the four tasks of the six subjects. The full analyses of mean and peak ratios are illustrated in the Appendix A (with the mean and peak values of the two ratios and ratios averaged to the mean ratio of the subject and the summed and absolute summed value for each subject).

All subjects were followed with radiographs over the medium term to maturity as part of the clinical follow-up. Subject 6 had a Cobb angle less than five degrees, thus improved to a straight spine. Subject 1 deteriorated with immediate worsening of the curve and had surgery shortly after testing. The three other subjects had an unchanged curve until maturity.

### 3.2. Electric Stimulation

Twenty-four patients were referred as subjects, but nineteen declined participation. We performed one session of electric stimulation on the five included subjects. The gender ratio (female:male) was 1:4 and the median age was 15.64 years of age (range: 2.82, Q1: 14.19, Q3: 16.45). The median Cobb angle at inclusion was 12.5 (range: 11.3, Q1: 11.3, Q3: 19.39). Table 1 (top) displays patient demographics, curve type and the stimulation amplitude.

The electric stimulations were performed uni- and bilaterally, at the level of or under the primary curve on the convex, concave and both sides of the primary curve as seen in Figure 4.

The cinematic sequences of radiographs and radiographs at maximal stimulation are shown in Appendix A. Table 2 (middle and lower) displays the calibrated changes in vertebral body displacement, angulation and various ratios of these at maximal stimulation.

## 4. Discussion

The purpose of this study was to investigate in AIS patients the feasibility of EMG evaluation during selected motor tasks and electric stimulation of AIS to examine the scoliogenic role of the spinal muscles. The sample size is small, and the findings consequently have limited generalizability. Evaluations with TES and EMG were performed once cross-sectionally and did not capture the dynamic nature of the disease progression of AIS. Therefore, our findings should be considered as suggestive and hypothesis-generating.

### 4.1. EMG

In this study, we selected two isokinetic tasks of left and right lateral bending and rotation when standing, since asymmetric muscle activity was expected [44]. As early as 1955, Riddle and Roaf [16] suggested investigating the EMG activity for scoliosis and Cheung et al. [48] suggested evaluating EMG as a predictor of risk of progression. In this study, we anticipated differences in muscle activity across the deformity, since previous studies have demonstrated asymmetry of the convex and concave muscles in AIS when compared to normal spines in lateral bending. It is especially interesting if this would be predictive for the development and determining risk of progression of AIS [21,28,44]. When evaluating local mean and peak EMG signals at the three levels, we were unable to find consistent differences in general. In some instances, convex activation was higher than concave—in particular at the apex level of the primary curve and for subjects with curve progression—as demonstrated in previous studies. However, this is an inconsistent finding [28]. When evaluating for differences in the mean and peak EMG signal ratios, we found differences between the concave and convex sides for a specific motor task of a specific subject but again without a consistent pattern emerging across the subjects, similar to previous studies [28]. However, following the path of EMG activity of progressive and non-progressive AIS, we had two consistent findings. One of the five subjects had rapid progression and one regressed to a straight spine. When evaluating the subjects by adding their absolute EMG ratios for all four motor tasks, the subject with progression had an almost 10-fold less summed mean and peak EMG ratio (X + Y) when compared to the stationary curves and the subject with regression. The latter had a more than 3-fold higher summed mean and peak EMG ratio (X/Y). We interpret the higher ratios as heightened asymmetric spinal muscle activity trying to rebalance the spine to maintain or correct the deformity. When progression occurs, this is preceded or accompanied by a decreased summated ratio across the deformity. Notice that our two mentioned subjects were diagnosed with cerebral palsy, which can affect muscle activity registered by EMG. However, due to their typical AIS curve and chronological age over 14, they may be characterized as AIS cases.

As mentioned, asymmetry in recorded EMG has been an inconsistent finding when utilized to detect the risk of curve progression in the literature [28]. Our findings suggest that rather than detecting asymmetric EMG patterns, we should identify a ‘silencing’ of summed mean and peak asymmetry. When monitored longitudinally, and when curve progression occurs, or preceding the curve progression, we should investigate a decrease in EMG asymmetry (or maybe less normalized EMG activity). Farahpour et al. [44] found that trunk (spine) muscle activities in AIS are asymmetrical and higher than those in normal subjects in AIS. The asymmetry in muscle response was interpreted as perturbations as a compensatory strategy in trying to rebalance the spine and not an inherent characteristic [9,44]. Muscle activity varies with the motor tasks in AIS [23] and asymmetrical muscle activity is induced during anticipatory postural perturbations and symmetrical muscle activity to sudden balance threats [49]. The asymmetrical muscle activity is suggested as a neuromuscular adaptation to the altered mechanical situation of AIS [44]. Based on these findings and our results, we suggest that asymmetric muscle activation is a postural neuromuscular adaptation when the spine rebalances itself towards the midline. Previous studies found that symmetric muscle activity is present in smaller AIS where less mechanical correction is needed [50,51,52], whereas in larger curve AIS, more corrective, asymmetric muscle hyper-activation is required [53,54]. Muscle endurance in AIS is similar to normal subjects initially, but with increased muscle activity and fatigue and there is an increase in convex rotational mobility without additional active postural stability in AIS [55]. Moreover, there seems to be a higher metabolic cost in AIS, thus a higher risk of fatigue [56]. This is indicative of the EMG ratios being affected or diminished as seen in our study. We suggest that asymmetric EMG activity will be detected as a compensatory correcting mechanism as a consequence of neuromuscular efficacy [24,57,58,59]. Approaching fatigue (and initially) when muscles are weakened, the (asymmetric) muscle activity will increase [54], but when the mechanical demands are superseded and following fatigue, the asymmetrical EMG decreases, rotational instability increases, and opens the path for a vicious cycle progression [10,14,55,60,61]. Interestingly, when examining differences in muscle strength for AIS, there is a consistent asymmetric rotation strength deficit and less relative trunk strength with a rotational weakness on the concave side of the primary curve in AIS [20,21,24,25].

### 4.2. Electric Stimulation

In this study, we performed unilateral or bilateral electric stimulation on subjects with small-curve AIS. At the initial visual evaluation, the frontal plane spatial changes in the vertebral bodies when stimulated only showed discrete changes without any apparent pattern. However, when analyzing the spatial changes with calibrated measurements of displacement and rotation (Cobb angle), we detected an overall average general compression of the vertebral bodies of 4.4 mm compression and 0.9° increase in CA, thus a 4% (Equation (3)) and 70% (Equation (4)) increase in CA, respectively, by paravertebral muscular stimulation. In the small deformity AIS, subject number 5 with a small curve (~13°) demonstrated a relatively large compressive effect in the relative ratio of CA/displacement (Equation (5)) with a ratio less than 1 (=0.61) when the spinal muscles were stimulated. In the deformity, with more than 20° of CA, subject 1 with a larger curve (~24°) demonstrated a relatively large increase in the relative ratio of CA/displacement (Equation (5)) exceeding four times (=4.33) when the spinal muscles were stimulated. In deformities less than 7° of CA, we found a 4.8 mm compression or expansion and 1.8° increase in CA, thus a 1% (Equation (3)) and 20% (Equation (4)) increase in CA. Based on this finding, we suggest that in small-curve AIS (CA 10–20°), the spinal muscles do not exert a scoliogenic response but rather have a stabilizing function to maintain the spine in the midline without a major effect/change in CA. Subsequently, the role of the muscles converts from counteracting and stabilizing the spine to being scoliogenic for curves with a CA of more than 20°. Interestingly, there was a vivid and diverse response in both displacement and angle for straight spines, which we interpret as spinal muscle alertness and muscle fatigue in AIS as discussed in the following section. For a deformity exceeding 20° CA, we induced a muscle response with moderate and unilateral electric stimulation (~55) on the concave side of the primary curve in a (the only) female subject at age 14.7. For a small deformity (10–20° CA), we induced a muscle response with a small and unilateral electric stimulation (~34) on the concave side but below the primary curve in a male subject at age 16.5. For a straight spine (CA < 10), we induced a muscle response with both large and small relative stimulation with electrodes placed both bi- and unilaterally in male subjects in the age range 13.7–16.4. Previous studies such as Bruggi et al.’s work [57] found that the spinal muscles maintained static and dynamic stability of the spine, thus achieving a corrective effect of a lumbar scoliotic curve. This corrective effect in small-curve AIS was affirmed by inducing a temporary paralysis of the iliopsoas leading to curve correction and derotation [32]. We were unable to identify studies using a similar experimental muscular stimulation as ours; thus, we are unable to relate our findings directly, but lateral electric surface stimulation (LESS) mediates correction after short-duration stimulation for initial scoliosis angles less than 25 ° [34], whereas for curves more than 30–40 °, it generates a shorter term correction [36,37]. This corroborates our findings indirectly. In experimental animal models [34,62], unilateral LESS led to a scoliosis-like deformity with the convexity located on the opposite side to the stimulated side. The extent of the deformity was ascribed to fatigue of stimulated spinal muscles by long-term serial stimulation, which is indirectly consistent with our EMG findings. LESS also accommodates changes in muscle fibres with a conversion to type 1 (fast) in the stimulated muscles [34,63]. This conversion follows histological findings of a significantly lower proportion of type I (slow-twitch oxidative) fibres in the spinal muscles on the concave side of the scoliotic curves in AIS, where the muscles adopt a faster or more glycolytic profile to accommodate reduced low-level tonic activity following curve progression or due to general disuse of the spinal muscles associated with trying to maintain or balance the spinal deformity [27,63]. Muscle fatigue characteristics are similar to normal patients; thus, this fibre conversion is reactive and a physiological response to the increased mechanical demand [50,55]. Long-term LESS seems to mediate compensatory changes in deformity, histology, number of motor endplates and muscle fibre type and long-term LESS seems to lead to exhaustion with a reduced number of motor endplates and histological atrophy [62,64,65]. We speculate if such gradual changes occur in the natural history of AIS with a decline in force level generation as a physiological response dependent on the duration and extent of central and peripheral fatigue and changing from being reversible to irreversible over time [66,67].

Noticeably, subject ‘1’ was the only female adolescent subject. Many of the potential subjects declined to participate either when informed of the experimental set-up with electric stimulation or when preparing the evaluation, i.e., in the process of applying the electrodes before stimulation. Certainly, the electric stimulation induced a pinprick-like sub-painful to minor painful sensation. The authors tested the procedure themselves before initiating the study. During an interim evaluation, we abandoned this part of the protocol due to the high number of dropouts. The finding using electric stimulation is based on five evaluations with subtle changes by the electric stimulation and unsystematic electric stimulation; thus, our results should be interpreted as suggestive. Responses from the stimulation were achieved only when the electrodes were placed close to the midline and when the gradual intensity was close to maximal; thus, we interpreted this as both uni and bi-lateral stimulation led to bilateral overall muscular stimulation of all the spinal muscles whether the electrodes were placed uni- or bilaterally. The subjects had similar ages but differences in curve deformity and electrode placement; thus, we are unable to substantiate our conclusions quantitatively.

When evaluating the EMG data, we also excluded a large number of subjects due to not performing all the selected motor tasks, not performing the five repetitions or the quality of movements and/or the EMG recordings being inadequate. In general, we included subjects as a convenience sample with variance in curve morphology and level, and varied Cobb angles (but was small-curve AIS). Moreover, the subjects were tested at different times from diagnosis and when reaching maturity. All radiological examinations were performed as part of the subject’s clinical follow-up, justifying the induced radiation and some subjects had regressed to an almost straight spine when examined. Since the focus was on small-curve AIS, the risk of progression was small for our population [15]; thus, using EMG for prediction should be considered circumstantial for our population, and it would be unlikely to be able to establish a predictive relationship for asymmetry as expressed in EMG ratios for our subjects. However, this might also be obscured by the other above-mentioned factors and the small number of subjects. Still, we believe that our findings are important to communicate, since electrostimulation is unique and probably will not be performed in the future due to the tenacity of the method.

A pioneering experimental animal study on muscle imbalance by Schwartzmann and Miles [68] produced scoliosis by the unilateral release of the sacrospinal muscles, but release without imbalance maintained a straight spine [68]. Rotational muscle imbalance was already considered an ‘old and forgotten idea’ in 1955 when the unopposed activity of the spinal muscles was hypothesized to cause vertebral rotation and initiate AIS deformity [16,69]. These studies indicate that spinal muscular imbalance plays a role in scoliogency; with our study, we would like to reintroduce the spinal muscles as an important factor in scoliogency by suggesting two mechanisms. Firstly, spinal muscles mediate stability and counteract scoliosis in the straight spine and small-curve AIS but are scoliogenic in further curve progression (>~20 ° of CA). Specific muscle exercises that seem to reduce the risk of AIS progression are indicative of this role of the spinal muscles [52,70,71]. Secondly, in the process of maintaining spinal balance, fatigue and morphological changes can occur. If the spinal muscles are unable to maintain balance, then curve progression occurs, and this is preceded or accompanied by the subsidence of EMG asymmetry as a heralding sign. We suggest that rather than looking for asymmetry in EMG responses, the detection of reduced EMG asymmetry could be taken as a sign that the spinal muscles are fatigued and failing to maintain stability for the spinal deformity. When the initial stabilizing spinal muscle actions are exhausted, they will be unable to counteract the deformity and change their role to becoming scoliogenic. Proof of these two mechanisms requires larger prospective studies with longitudinal EMG monitoring, showing the suggested decrease in EMG ratios in deformity progression, and tests of interventions with a focus on specific rotational muscle strengthening for influence on the natural development of AIS [10].

Based on the findings, we suggest that future studies should entail larger scale and prospective studies. These studies should include more comprehensive EMG tests, such as being related to the various curve types and characteristics, performing different motor tasks, examining different EMG signals (onset and amplitude), systematic electrode positioning and relating this to curve progression. The application of electric stimulation to the spinal muscles for investigation of AIS as applied in this study is a novel technique. We applied the electric current with an increasing intensity until we detected the movement of the vertebral bodies in our radiographic sequence. We suggest that these techniques of electric stimulation should be investigated with the effect of various types of stimulation techniques concerning electrode placement and various applied intensities regarding strength, pulse duration and frequency. The effect of electric stimulation should be investigated in regard to the various curve types and curve severity (i.e., >20° in Cobb angle). The effects of muscle strengthening and bracing should be examined concerning EMG response to attain a deeper understanding. Prospective studies would also provide more robust evidence and establish proposed causal relationships. In conclusion, the role of spinal muscles in the development and progression of AIS suggests two key findings. In small-curve AIS, spinal muscles appear to stabilize the spine, whereas in larger curves (>20° Cobb A = angle), they have a scoliogenic effect, potentially contributing to curve progression. We suggest that the role of the muscles converts from counteracting AIS and stabilizing the spine to being scoliogenic. Moreover, we interpret higher EMG ratios as heightened asymmetric spinal muscle activity when the spinal muscles try to balance the spine to maintain or correct the deformity. When progression occurs, this is preceded or accompanied by decreased EMG ratios.

## Figures and Tables

**Figure 1 jcm-13-01758-f001:**
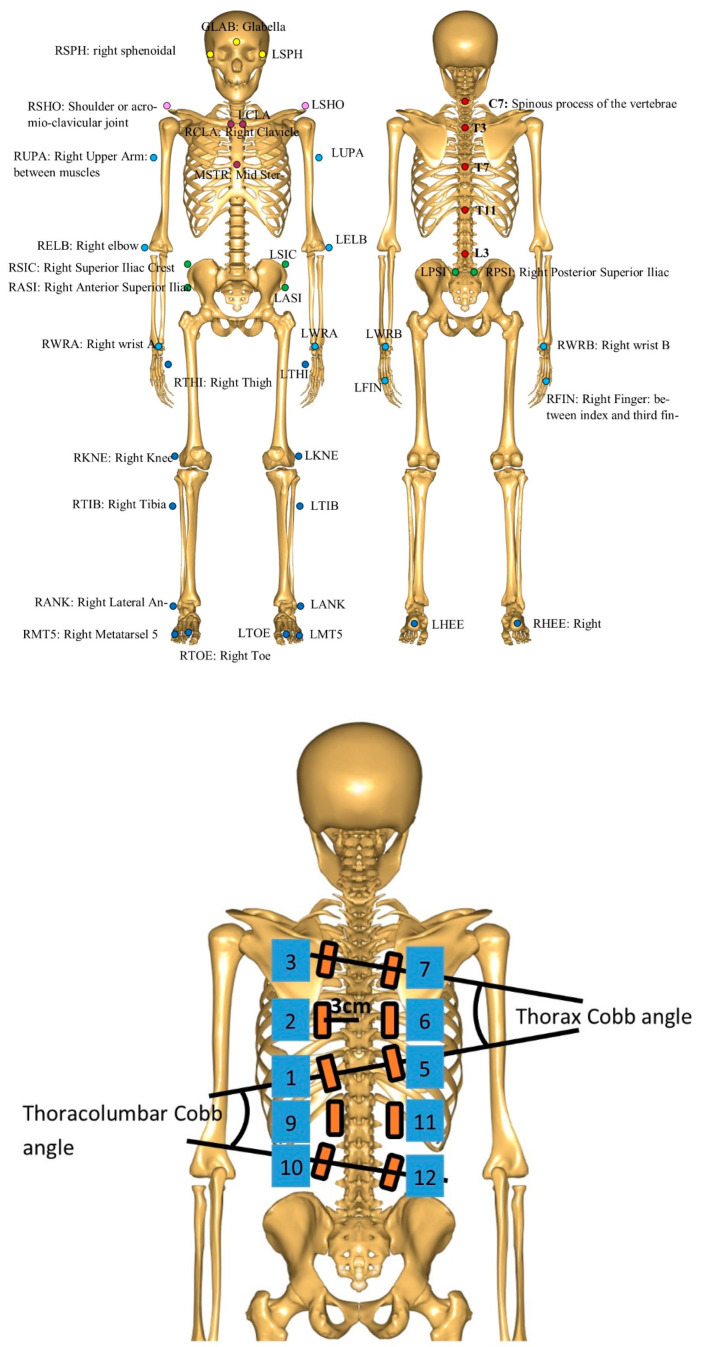
Top frame: The positioning of the 43 reflective markers on the skin over various bone projections on the head, back and pelvic bone, shoulder, and upper and lower limb. Bottom frame: The principal placement of the ten Myon electrodes on the back. This was tailored to the primary curve of the AIS.

**Figure 2 jcm-13-01758-f002:**
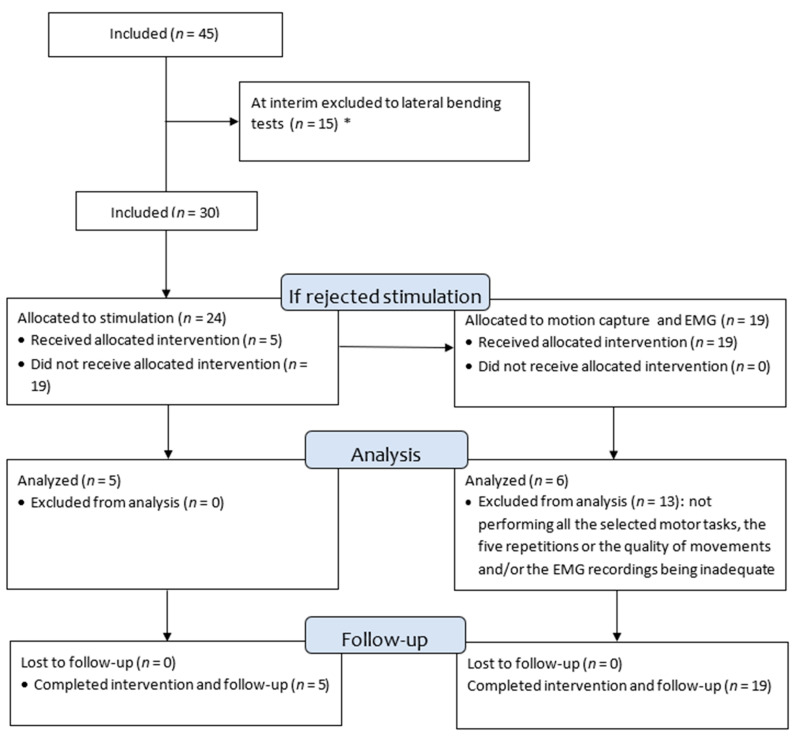
Timeline of the history of the subject inclusion. Of the 30 subjects, 24 declined to participate in electric stimulation and 19 were allocated to participate in motion capture and EMG recording, whereas 6 had completed a full set of EMG. * Following interim analysis, we closed the two arms of this study and allocated the 15 subjects to another AIS project of controlled lateral bending.

**Figure 3 jcm-13-01758-f003:**
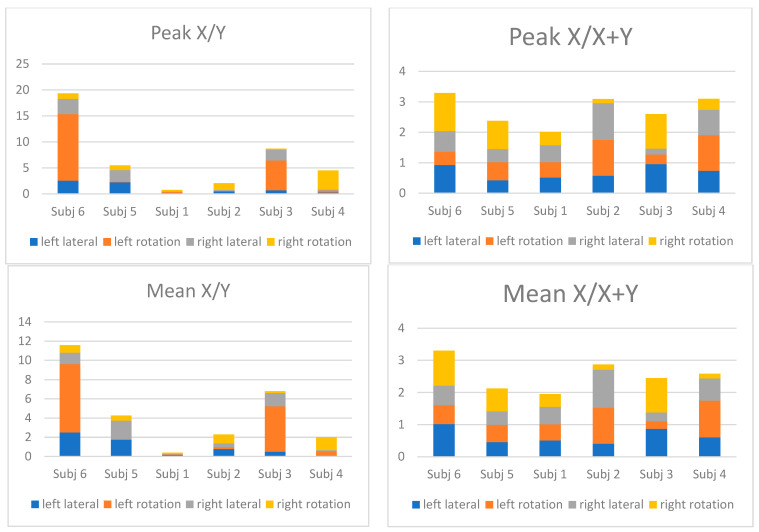
The summed absolute values for the mean and peak EMG ratios of Formulae (1) X/Y and Formulae (2) X/X + Y of the six subjects.

**Figure 4 jcm-13-01758-f004:**
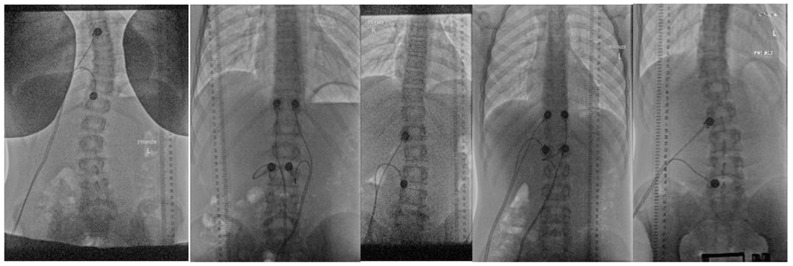
The electrode placement for subjects 1, 2, 3, 4 and 5 for electric stimulation.

**Table 1 jcm-13-01758-t001:** Patient data of gender and age, curve type and size, history of AIS, MRI findings, the amplitude of stimulation and electrode placement for electric stimulation (upper) and patient data of gender and age, curve type and change in Cobb angle with the initial value, at examination and maturation, MRI findings, history of AIS and medical history and surgical history and brace history and if referred to/participating in physiotherapy (PT = refers to physiotherapy, * = for back pain).

**Subject**	**Age at Examination**	**Sex**	**Curve Type**	**Cobb Angle**	**History**	**MRI**	**Stimulation Amplitude**	**Electrode Placement**	**Brace History and Physiotheraphy**
1	14.7	F	sinistrokonvex TL S	22.3	Stationary		55	unilateral concave	Soft brace and PT
2	16.4	M	dextrokonvex TL > straight	11.0	Regression		63 × 2	bilateral symmetrical	No brace and no PT
3	25.7	M	dexkonvex TL S	11.6	Stationary	Arcolysis L5/Spina bifida	94	unilateral concave	No brace and PT
4	13.7	M	sin. convex T > straight	12.5	Regression		62 × 2	bilateral symmetrical	No brace and no PT
5	16.5	M	dextrokonvex TL S	13.3	Stationary		94	lower convex	No brace and PT *
**Subject**	**Age at Examination**	**Sex**	**Curve Type**	**Cobb Angle**	**History**	**MRI**	**Medical History**	**Surgical AIS**	**Brace History and Physiotheraphy**
1	14.9	F	Low dextrokonvex T	18 > 56 > 0	Progression	Synrinx	CP	Operated shorthly after	Soft brace and PT
2	14.7	F	Sinkonvex TL	19 > 26 > 21	Regression	none	TDC	None	Soft brace and PT
3	17.8	F	Low dextrokonvex T	26 > 19 > 17	Regression	ia	TDC	None	No brace and PT *
4	12	F	Dextro konvex T	13 > 11 > 9	Regression	none	TDC	None	No brace and no PT
5	12.8	F	Dextrokonvex TL	11	Stationary		TDC	None	No brace and PT
6	14.3	F	Dextrokonvex TL	15 > 15 > 0	straight		CP	None	Soft brace and PT

**Table 2 jcm-13-01758-t002:** Changes in calibrated vertebral body displacement and rotation by stimulation in absolute values for each subject (upper) and when evaluated for specific ranges of Cobb’s angle; the changes in vertebral body distances (Equation (3)) and Cobb’s angles (Equation (4)) when stimulated and the ratio when divided by deformity magnitude and when evaluated for not (un-)normalized (not normalized to the unstimulated distance or angle) and normalized values (Equation (5)) (lower). * For secondary curve.

	**du**	**ds**	**αu**	**αs**	***du**	***ds**	***αu**	***αs**
Subject 1	137.2	136.4	26.5	28.8				
Subject 2	150.9	128.9	1.3	4				
Subject 3	88.3	87.4	5.7	6.3				
Subject 4	73.8	70.7	2.8	1.4				
Subject 5	138.7	106.9	10.9	11.7	112.4	105.3	18.8	21.5
**CA**	**R** **dis**	**R** **ang**	**R** **un**	**R** **n**				
<6°	0.066	−0.894	0.414	13.55				
~10°	0.23	−0.073	0.025	0.32				
>20 °	0.006	−0.087	2.875	14.47				

## Data Availability

For further interest, inquiries for data can be sent to the corresponding author.

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
