# Peer review of "Using Electric Stimulation of the Spinal Muscles and Electromyography during Motor Tasks for Evaluation of the Role in Development and Progression of Adolescent Idiopathic Scoliosis"

_jcm, 2024, doi:10.3390/jcm13061758_

Round 1
Reviewer 1 Report
Comments and Suggestions for Authors
My review is as below
1. The study titled 'Using Electric Stimulation of the Spinal Muscles and Electromyography during Motor Tasks for Evaluation of the Role in Development and Progression of Adolescent Idiopathic Scoliosis' is found valuable.
2. Participants with a Cobb angle between 10-30 degrees were included. How many of these participants received physiotherapy and/or brace treatment during this period? It would be good if it was stated which cases were those who received physiotherapy and/or brace treatment among those who received EMG and electrical stimulation.
3. ‘Asymmetric muscle activation is a postural neuromuscular adaptation with muscle activity/EMG perturbations when the spine balances itself towards the midline’ point is valuable.
4. Even if a nice study, larger prospective studies are needed.
5. There is a need for muscle strengthening and the results and/or effects of brace use in participants with a Cobb angle over 20 degrees.
Author Response
We thank the reviewer for this insightful review.
- The study titled 'Using Electric Stimulation of the Spinal Muscles and Electromyography during Motor Tasks for Evaluation of the Role in Development and Progression of Adolescent Idiopathic Scoliosis' is found valuable.
We thank the reviewer for his/her point of view. Thank you.
- Participants with a Cobb angle between 10-30 degrees were included. How many of these participants received physiotherapy and/or brace treatment during this period? It would be good if it was stated which cases were those who received physiotherapy and/or brace treatment among those who received EMG and electrical stimulation.
This is very interesting information, and physiotherapy would i.e. strengthen the muscles and affect EMG. We have added the brace history and referral/participation in physiotherapy to Table 1. We did not formally monitor their brace use and participation in spine exercises/physiotherapy, so we have denoted this as ‘referral to/participation in physiotherapy’.
-‘Asymmetric muscle activation is a postural neuromuscular adaptation with muscle activity/EMG perturbations when the spine balances itself towards the midline’ point is valuable.
We thank the reviewer for sharing this point of view.
- Even if a nice study, larger prospective studies are needed.
and
- There is a need for muscle strengthening and the results and/or effects of brace use in participants with a Cobb angle over 20 degrees.
We absolutely agree with the reviewer. In this study, we were unable to provide this, but have added this for proposed new studies of this type:
We agree with the reviewer. We write in l. 499:
…Proof of the validity of these two proposed mechanisms will first arise after larger prospective studies with longitudinal EMG monitoring, showing the suggested decrease in EMG ratios in deformity progression…
We have added to the Discussion:
Statistical analyses were performed using IBM SPSS Statistics, Version 25 (IBM, Richmond, VA, USA) using descriptive statistics for determining medians and quartiles of the age and Cobb angle. Further statistical analyses for significance were not carried out due to the low sample size, thereby limiting the accuracy and generalizability of study results with reduced reliability, a higher margin of error, lower statistical power, increased variability, risk of bias and higher risk of type I and II errors. (l.254-l.259) .
And
The sample size is small, and findings consequently have limited generalizability. Evalua-tions with TES and EMG were performed once cross-sectionally and did not capture the dynamic nature of the disease progression of AIS. Therefore, our findings should be con-sidered as suggestive and hypothesis-generating. (l.331-l.334)
And
Based on the findings, we suggest that future studies should entail larger-scale and prospective studies. These studies should include more comprehensive EMG tests, such as being related to the curve characteristic, performing several Motor Task types, EMG signal (onset and amplitude), electrode positioning and relation to curve type and curve progression status. For Electric Stimulation to incorporate a uniform stimulation technique and additionally, stratification to the various curve types and curve severity (i.e. > 20 ° in Cobb Angle). The effects of muscle strengthening and brace use should be examined concerning EMG response to attain a deeper understanding. Prospective studies would also provide more robust evidence and establish proposed causal relationships. (l. 504-l. 512)
Reviewer 2 Report
Comments and Suggestions for Authors
- authors should choose either American (etiology) or British (aetiology) English
- please add a Conclusions section to the manuscript with 2-3 phrases concluding your work
- "dependant" in "muscle-dependant" is incorrect
- assumption that larger curves will progress - while it is generally accepted that larger curves have a higher risk of progression - the statement could be interpreted as deterministic - please revise
- briefly acknowledge the current standard practices for prediction and management of AIS
- which muscle imbalance influences AIS development and progression? please concisely introduce this
- inconsistent capitalization of letters in terms like "Electric Stimulation" and "Motor Tasks."
- just a comment - and please acknowledge this as a limitation - Cross-Sectional Study Design - using this design to investigate AIS that evolves over time might not capture the dynamic nature of the disease progression effectively
- include a brief description of the statistical tests used, why they were selected, and how they were applied to your specific data
- something is unclear - how was information presented to ensure that adolescent participants and their caregivers fully understood the study purpose, procedures, risks, and benefits
- "diagnozed" should be "diagnosed"
- expand on the limitations of the small sample size and how they affect your study
- at the end, please consider to outline specific, actionable research questions/ hypotheses generated by the your findings
Comments on the Quality of English Language
I included some error examples in my comments. However, English should be properly re-reviewed before publication.
Author Response
We thank the reviewer for this insightful review.
- authors should choose either American (etiology) or British (aetiology) English
We have changed the use of aetiology to the American ‘etiology’ in l. 54 and l. 61
- please add a Conclusions section to the manuscript with 2-3 phrases concluding your work
We have added:
. In conclusion, the role of spinal muscles in the development and progression of AIS sug-gests two key findings. In small curve AIS, spinal muscles appear to stabilize the spine, whereas in larger curves (> 20° Cobb Angle) they have a scoliogenic effect, potentially con-tributing to curve progression. We suggest that the role of the muscles converts from coun-teracting AIS and stabilizing the spine to being scoliogenic. Moreover, we interpret higher EMG ratios as heightened asymmetric spinal muscle activity when the spinal muscles try to balance the spine to maintain or correct the deformity. When progression occurs, this is preceded or accompanied by decreased EMG ratios.. (l. 518- l. 526)
- "dependant" in "muscle-dependant" is incorrect
This is changed in l. 63 and 448 to ‘dependent’
- assumption that larger curves will progress - while it is generally accepted that larger curves have a higher risk of progression - the statement could be interpreted as deterministic - please revise
We agree with the reviewer and have modified and changed ‘progression’ to ‘risk of progression’ in l. 95, l. 316, l. 320, l. 342, l. 471
We have distinguished between if there is perceived that if an EMG change/decrease, that would change the risk of progression (then changed the phrase ‘progression’ to ‘risk of progression’), or if there is described as an occurring/ongoing curve progression (then not changed the phrase ‘progression’ to ‘risk of progression’).
- briefly acknowledge the current standard practices for prediction and management of AIS
We have added …
1.3. The Current Standard Practices for AIS Prediction and Treatment
The current standard practices for predicting and managing AIS focus on early detec-tion, monitoring, and intervention based on the severity and progression risk of the cur-vature. For mild curves, observation and regular monitoring of the spine's curvature to track curve progression is common with periodic clinical evaluations and radiographic assessments, where the risk of progression can be evaluated by a combination of serial X-rays, menarche history, serial height measurement, triradiate cartilage status, Risser grade, and Sanders skeletal maturation staging. In moderate curves in the growing ado-lescent, bracing is often utilized. Both observation and bracing can be combined with physiotherapy and exercises, such as the Schroth method. Surgical interventions are offered in severe cases of AIS when rapidly progressing or causing significant problems such as back pain.. (l. 108-l.119)
- which muscle imbalance influences AIS development and progression? please concisely introduce this
This is a very good question. I’m not sure that we are able to determine this yet. In the current study, we believe that we stimulated all (many) of the spinal muscles between/surrounding the electrodes, thus it was the sum of muscle ‘forces’ affecting the vertebral bodies as seen in the cinematic sequences. In a previous literature previous study, we mapped the spinal muscles' anatomical location and function from the existing literature:
Hansen , de Zee M, , Andersen TB, Wong C, Rasmussen J, Simonsen EB. Anatomy and biomechanics of the lumbar spine with special reference to biomechanical modelling. (Spine. 31(17):1888-1899, August 1, 2006).
And in my publication: Wong C. Mechanism of Idiopathic Scoliosis; a Unifying Theory of Development by Normal Growth and Imbalance. (Scoliosis 2015, 10:2.), I hypnotized that ...treatment entails correction of the position of the thoracic sagittal curvature as well as the stretching of thoracic ligaments and strengthening rotational thoracic and lumbar muscles…so I believe loss of function of the muscles providing rotational thoracic and lumbar stability is essential.
In my previous publication ‘Wong C, Gosvig K, Sonne-Holm S. Testning temporary paralysis of the iliopsoas muscle as a treatment in idiopathic adolescent scoliosis. (Scoliosis Spinal Disord. 2017; 12: 33)’, I evaluated the psoas muscle for ‘a hypnotized scoliogenic role of the Psoas muscle would be that of initiating or maintaining a lumbar scoliotic curvature by muscle contraction’ using botulinum toxin injections. We found a significant improvement (lesser curve) in thoracic and lumbar Cobb’s angle and a non-significant thoracic and significant lumbar derotation (changes in Nash and Moe’s classification), and a non-significant small average change in rib vertebra angles with an improvement on the convex side and a deterioration on the concave side. We interpreted this as that the spine muscles do play a role in maintaining the human adolescent idiopathic scoliosis by the muscle contraction or pull by the Psoas muscle, which was to be expected if the muscle pull by contraction was released in the lumbar area with subsequent effect in the thoracic area.
Griva et al (2016) suggested a pathomechanic role for quadrates lumborum as affecting lateral lumbar curves (LLC) in ‘Grivas TB, Burwell RG, Kechagias V, Mazioti C, Fountas A, Kolovou D, Christodoulou E. Idiopathic and normal lateral lumbar curves: muscle effects interpreted by 12th rib length asymmetry with pathomechanic implications for lumbar idiopathic scoliosis. Scoliosis Spinal Disord. 2016 Oct 14;11(Suppl 2):35. doi: 10.1186/s13013-016-0093-8. PMID: 27785474; PMCID: PMC5073422.’
I think that we will be able to determine this using various types of simulation as inverse and forward dynamic and finite element analysis. We started using this many years ago:
de Zee M, Hansen L, Andersen TB, Wong C, Rasmussen J, Simonsen EB. A Generic Detailed Rigid-body Lumbar Spine Model. (J Biomechanics, 40,1219-1227, 2007).
Wong C, Erik Simonsen, de Zee M, Rasmussen J, Hansen L, Dendorfer S. The effects of muscles in the Lumbar Spine. (Open Spine Journal, Volume 3, 2011, 21-26)
And in a more recent publication, these models are elaborated and might answer your question:
Hamed Shayestehpour, John Rasmussen, Pavel Galibarov, and Christian Wong. An articulated spine and ribcage kinematic model for simulation of scoliosis deformities (Multibody Syst Dyn 53, 115–134 (2021). https://doi.org/10.1007/s11044-021-09787-9)
We have added:
1.4. The Role of Muscle Imbalance in the Development and Progression of AIS
Muscle imbalances are hypothesized to influence the development and progression of AIS [10,14,16], involving primarily the muscles that provide rotational stability to the thoracic and lumbar regions, as well as specific muscles such as the psoas and quadratus lumborum. Wong [10] suggested spinal muscles with thoracic and lumbar rotational functions, thus maintaining ‘rotational’ stability to those areas are key determinants in a ‘common pathway of imbalance’ in AISand loss of function of those muscles would facilitate AIS. Previous publications by Wong et al. [29] and Griva et al. [32] showed that an intervention targeting the psoas muscle by injections of botulinum toxin resulted in improvements in the curvature and rotation of the spine, and radiological evaluation of asymmetry of the 12th rib length suggested that the quadratus lumborum muscle could affect lateral lumbar curves in AIS. (l.121-l.132)
- inconsistent capitalization of letters in terms like "Electric Stimulation" and "Motor Tasks."
We have now changed the terms "electric stimulation" and "motor tasks" with consistent capitalization. Except for the subheadings, where these have been fully capitalized throughout the manuscript.
- just a comment - and please acknowledge this as a limitation - Cross-Sectional Study Design - using this design to investigate AIS that evolves over time might not capture the dynamic nature of the disease progression effectively
We agree with the reviewer and have added:
... . The sample size is small, and findings consequently have limited generalizability. Evalua-tions with TES and EMG were performed once cross-sectionally and did not capture the dynamic nature of the disease progression of AIS. Therefore, our findings should be con-sidered as suggestive and hypothesis-generating.... (l. 330-l. 333)
- include a brief description of the statistical tests used, why they were selected, and how they were applied to your specific data:
We have added:
Statistical analyses were performed using IBM SPSS Statistics, Version 25 (IBM, Richmond, VA, USA) using descriptive statistics for determining medians and quartiles of the age and Cobb angle. Further statistical analyses for significance were not carried out due to the low sample size, thereby limiting the accuracy and generalizability of study results with reduced reliability, a higher margin of error, lower statistical power, increased variability, risk of bias and higher risk of type I and II errors. (l.253-l.258)
- something is unclear - how was information presented to ensure that adolescent participants and their caregivers fully understood the study purpose, procedures, risks, and benefits
This followed the local national guidelines for first written information and then oral information to achieve informed consent. This was performed by an experienced pediatric orthopedic surgeon to ensure the subject and their caregivers understood the study purpose, procedures, risks, and benefits. We emphasized that the subjects (and parents) understood the immediate discomfort of electric stimulation. All participating researchers tried this themselves. However, some subjects did not want to participate in electric stimulation just before participating, and in retrospect, we should have let them try electric stimulation on i.e. their finger when informing them.
We have added:
Forty-five patients with their caregivers were included as subjects after receiving oral and formal written information by an experienced pediatric orthopedic surgeon using the national guidelines for informed consent and to ensure the subject and their caregivers understood the study purpose, procedures, risks, and benefits. (l. 261- l. 265)
And as described in the manuscript:
We obtained oral and written consent from the caregivers and the older subjects (when appropriate), and the study was conducted according to national guidelines and the Helsinki Declaration. (l.135-l.138)
- "diagnozed" should be "diagnosed"
We have changed and are using the "diagnosed" in l.114 and l. 340.
- expand on the limitations of the small sample size and how they affect your study
We have added:
…In general, our sample is small and findings will as a consequence limit the accuracy and generalizability of study results with reduced reliability, a higher margin of error, lower statistical power, increased variability, risk of bias and higher risk of type I and II errors, and evaluations with TES and EMG were performed once cross-sectionally, thus might not capture the dynamic nature of the disease progression of AIS effectively. Therefore, our conclusions should be considered as suggestive and hypothesis-generating. (l. 255-l. 258)
- at the end, please consider to outline specific, actionable research questions/ hypotheses generated by the your findings
We have added:
Based on the findings, we suggest that future studies should entail larger-scale and prospective studies. These studies should include more comprehensive EMG tests, such as being related to the curve characteristic, performing several Motor Task types, EMG signal (onset and amplitude), electrode positioning and relation to curve type and curve progression status. For Electric Stimulation to incorporate a uniform stimulation technique and additionally, stratification to the various curve types and curve severity (i.e. > 20 ° in Cobb Angle). The effects of muscle strengthening and brace use should be examined concerning EMG response to attain a deeper understanding. (l.503-l. 510)
Comments on the Quality of English Language
I included some error examples in my comments. However, English should be properly re-reviewed before publication.
We have asked an English speaker with a significant scientific background to revise the manuscript.
Round 2
Reviewer 2 Report
Comments and Suggestions for Authors
I agree with the changes.